# Inductive Biases for Object-Centric Representations
# in the Presence of Complex Textures

**Samuele Papa**[*1]         **Ole Winther**[2,3]         **Andrea Dittadi**[*2,4]

[1]POP-AART Lab, University of Amsterdam & The Netherlands Cancer Institute
[2]Technical University of Denmark
[3]University of Copenhagen & Copenhagen University Hospital
[4]Max Planck Institute for Intelligent Systems, Tübingen, Germany

## Abstract

Understanding which inductive biases could be helpful for the unsupervised learning of object-centric representations of natural scenes is challenging. In this paper, we systematically investigate the performance of two models on datasets where neural style transfer was used to obtain objects with complex textures while still retaining ground-truth annotations. We find that by using a single module to reconstruct both the shape and visual appearance of each object, the model learns more useful representations and achieves better object separation. In addition, we observe that adjusting the latent space size is insufficient to improve segmentation performance. Finally, the downstream usefulness of the representations is significantly more strongly correlated with segmentation quality than with reconstruction accuracy.

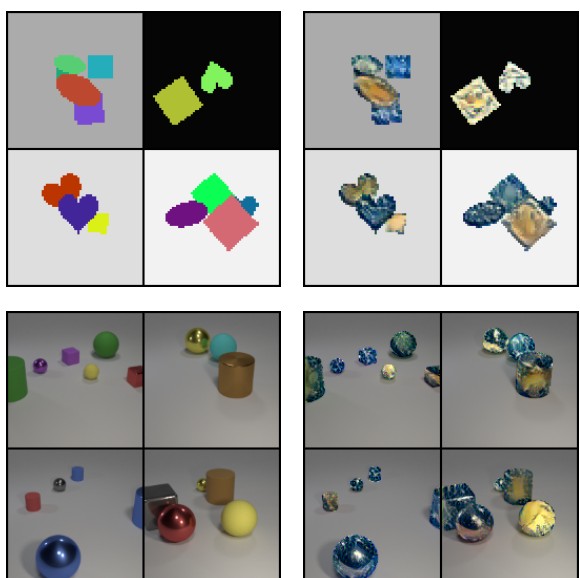

Figure 1: Left: samples from the original datasets. Right: samples from the same datasets with neural style transfer.

## 1   INTRODUCTION

A core motivation for object-centric learning is that humans interpret complex environments such as natural scenes as the composition of distinct interacting objects. Evidence for this claim can be found in cognitive psychology and neuroscience (Spelke, 1990; Téglás et al., 2011; Wagemans, 2015), particularly in infants (Dehaene, 2020, Chapter 3). Additionally, these concepts have already been successfully applied, e.g., to reinforcement learning (Berner et al., 2019; Mambelli et al., 2022; Vinyals et al., 2019) and physical modelling (Battaglia et al., 2016; Sanchez-Gonzalez et al., 2020). Current object-centric learning approaches try to merge the advantages of connectionist and symbolic methods by representing each object with a distinct vector (Greff et al., 2020). The problem of object separation becomes central for unsupervised methods that can only use the data

itself to lean how to isolate objects. Several methods have been proposed to provide inductive biases to achieve this objective (e.g., Burgess et al., 2019; Engelcke et al., 2020a,b; Kipf et al., 2021; Locatello et al., 2020). However, they are typically tested on simple datasets where objects show little variability in their texture, often being monochromatic. This characteristic may allow models to successfully separate objects by relying on low-level characteristics, such as color (Greff et al., 2019), over more desirable high-level ones, such as shape.

Research in the direction of natural objects is still scarce (Engelcke et al., 2021; Karazija et al., 2021; Kipf et al., 2021), as such datasets often do not provide exhaustive knowledge of the factors of variation, which are very rich in natural scenes. In this context, unsupervised methods struggle to learn object-centric representations, and the reason for this remains unexplained (Greff et al., 2019, Section 5).

---

[*]Correspondence to <s.papa@uva.nl> and <adit@dtu.dk>.

*Accepted for the Causal Representation Learning workshop at the 38[th] Conference on Uncertainty in Artificial Intelligence* (UAI CRL 2022).

*In this paper, we conduct a systematic experimental study on the inductive biases necessary to learn object-centric representations when objects have complex textures.* To obtain significant and interpretable results, we focus on static images and use *neural style transfer* (Gatys et al., 2016) to apply complex textures to the objects of the Multi-dSprites (Kabra et al., 2019) and CLEVR (Johnson et al., 2017) datasets, as shown in Fig. 1. The increase in complexity is, therefore, controlled. On the one hand, we still have all of the advantages of a procedurally generated dataset, with knowledge over the characteristics of each object, thus avoiding the above-mentioned pitfalls of natural datasets. On the other hand, we present a much more challenging task for the models than the type of data commonly used in unsupervised object-centric learning research.

We investigate MONet (Burgess et al., 2019) and Slot Attention (Locatello et al., 2020), two popular slot-based autoencoder models that learn to represent objects separately and in a common format. The latter obtains object representations by applying Slot Attention to a convolutional embedding of the input, and then decodes each representation into shape and visual appearance via a *single* component. In contrast, MONet reconstructs them with *two* components: a recurrent attention network that segments the input, and a variational autoencoder (VAE) (Kingma and Welling, 2014; Rezende et al., 2014) that separately learns a representation for each object by learning to reconstruct its shape and visual appearance. Unlike in Slot Attention, the shape reconstructed by the VAE is *not* used to reconstruct the final image; instead the shape from the recurrent attention network is used. To still have shape information in the latent representation of the VAE, the training loss includes a KL divergence between the mask predicted by the VAE and the one predicted by the attention network. Therefore shape and visual appearance are, in effect, *separate* unless the KL divergence provides a strong enough signal. For this study, we posit two desiderata for object-centric models, adapted from Dittadi et al. (2021b):

**Desideratum 1.** *Object separation and reconstruction.* The models should have the ability to accurately separate and reconstruct the objects in the input, even those with complex textures. For the models considered here, this means that they should correctly segment the objects and reproduce their properties in the reconstruction, including their texture.

**Desideratum 2.** *Object representation.* The models should capture and represent the fundamental properties of each object present in the input. When ground-truth properties are available for the objects, this can be evaluated via a downstream prediction task.

We summarize our **main findings** as follows:

1. Models that better balance the importance of both shape and visual appearance of the objects seem to be less prone to what we call *hyper-segmentation* (see

Section 3). We show how this can be achieved with an architecture that uses a single module to obtain both shape and visual appearance of each object. When this is not the case, it becomes significantly more challenging for a model to correctly separate objects and learn useful representations.

2. Hyper-segmentation of the input leads to the inability of the model to obtain useful representations. Separation is a strong indicator of representation quality.

3. The *representation bottleneck* is not sufficient to regulate a model's ability to segment the input. Tuning other hyperparameters such as encoder and decoder capacity appears to be necessary.

In the remainder of this paper, we will present the methods and experimental setup underlying our study, discuss our findings, and lay out practical suggestions for researchers in object-centric learning who might be interested in scaling these methods to natural images.

## 2 METHODS

In this section, we outline the elements of our study, highlighting the reasons behind our choices.

**Datasets.** Similarly to Dittadi et al. (2021b), we use neural style transfer (Gatys et al., 2016) to increase the complexity of the texture of the objects in the Multi-dSprites and CLEVR datasets (see Appendix B for details). This allows for textures that have high variability but are still correlated with the shape of the object, as opposed to preset patterns as done in Greff et al. (2019) and Karazija et al. (2021) or completely random ones. We apply neural style transfer to the entire image and then select the objects using the ground-truth segmentation masks (see Fig. 1). Keeping the background simple allows for a more straightforward interpretation of the models' performance.

**Models.** The models we study are MONet (Burgess et al., 2019) and Slot Attention (Locatello et al., 2020), that approach the problem of separation in two distinct ways, as mentioned in Section 1. MONet uses a recurrent attention module to compute the shape of the objects, and only later is this combined with the visual appearance computed by the VAE from the object representations. Instead, Slot Attention incorporates everything into a single component, with the shape and visual appearance of each object reconstructed from the respective object representation.

**Evaluation.** Following the two desiderata in Section 1, we separately focus on the *separation*, *reconstruction*, and *representation* performance of the models. *Separation* is measured by the Adjusted Rand Index (ARI) (Hubert and Arabi, 1985), which quantifies the similarity between two partitions

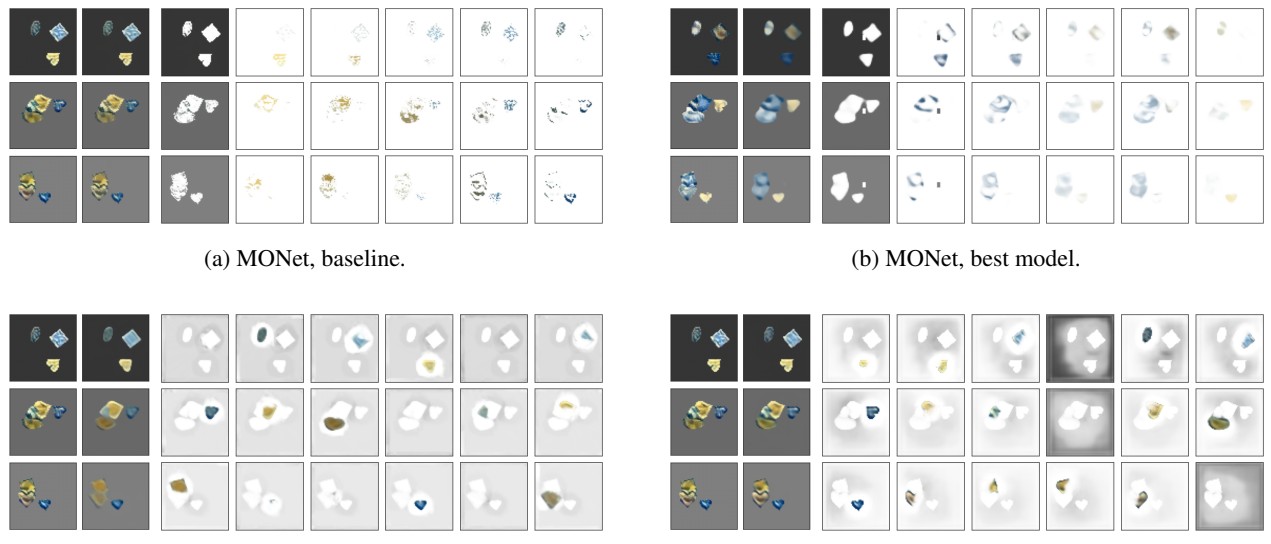

(a) MONet, baseline.

(b) MONet, best model.

(c) Slot Attention, baseline.

(d) Slot Attention, best model.

Figure 2: Reconstruction and separation performance on Multi-dSprites (from the validation set). From left to right in each subfigure: original input, final reconstruction, and product of the reconstruction and mask for each of the six slots. The improved architecture for Slot Attention splits objects less often. MONet still fails to separate objects correctly although it blurs the reconstructions. See Fig. 13 (Appendix G) for similar results on CLEVR.

of a set. The ARI is 0 when the two partitions are random and 1 when they are identical up to a permutation of the labels. *Reconstruction* is measured using the Mean Squared Error (MSE) between input and reconstructed images. *Representation* is measured by the performance of a simple downstream model trained to predict the properties of each object using only the object representations as inputs. Following previous literature (Dittadi et al., 2021b; Locatello et al., 2020), we match ground-truth objects with object representations such that the overall loss is minimized.

**Performance studies.** The *baseline* performance of the models on the style transfer datasets is established using the hyperparameters from the literature. We then vary parameters and architectures to improve performance. In MONet, we reduce the number of skip-connections of the U-Net in the attention module, we change the latent space size, the number of channels in the encoder and decoder of the VAE, and the $\beta$ and $\sigma$ parameters in the training objective. In Slot Attention, we increase the number of layers and channels in both encoder and decoder and increase the size of the latent space. For both models, we investigate how the latent space size alone affects performance. We use multiple random seeds to account for variability in performance when feasible (see Appendix E for further details).

## 3 EXPERIMENTS

In this section, we present and discuss the experimental results of our study. We first look at how different architec-

tural biases affect object separation. Then, we investigate representation performance with a downstream property prediction task. Finally, we study how the latent space size alone affects object separation and reconstruction quality.

**Architectural biases.** Qualitatively, the MONet baseline (Fig. 2a) segments primarily according to color, resulting in each slot encoding fragments of multiple objects that share the same color. We call this behavior *hypersegmentation*. On the other hand, the Slot Attention baseline (Fig. 2c) produces blurred reconstructions and avoids hypersegmentation. Here, some objects are still split across more than one slot but, unlike in MONet, we do not observe multiple objects that are far apart in the scene being (partially) modeled by the same slot. We observe this quantitatively in Fig. 4 (top): compared to the Slot Attention baseline, the MONet baseline has a significantly worse ARI score but a considerably better MSE. Fig. 13 in Appendix G shows similar results on CLEVR.

These observations can guide our search for better model parameters. Slot Attention is blurring away the small details of the texture and focuses on the shape to separate them. MONet, instead, achieves good reconstructions but does so by using the attention module to select pixels that share the same color, as opposed to entire objects, while the VAE simply reconstructs plain colors (see Appendix G for more details). Therefore, for MONet we attempt to sacrifice some reconstruction quality in exchange for better object separation. For Slot Attention, we investigate whether improving the reconstructions negatively affects object separation. We

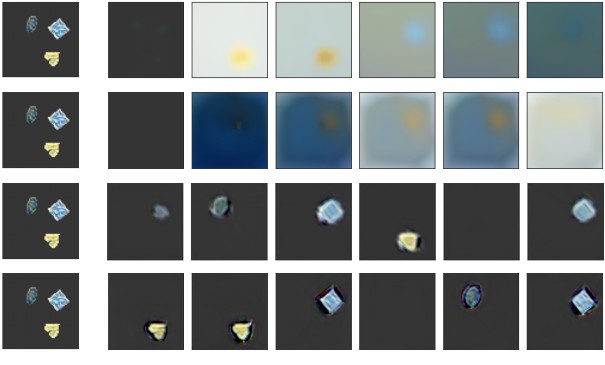

Input | Visual appearance reconstruction

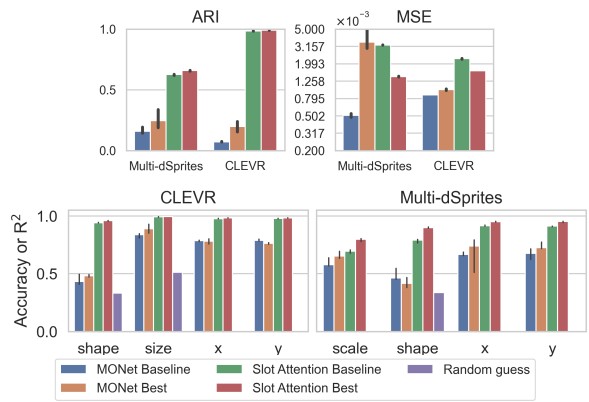

Figure 3: Qualitative results for the reconstruction of the visual appearance of the objects on Multi-dSprites (from the validation set). In each of the 4 groups, we show the input and the reconstruction of the visual appearance from each of the six slots, before masking. From the top: MONet Baseline, MONet Best Model, Slot Attention Baseline, Slot Attention Best Model. MONet primarily reconstructs plain colors with little to no regard to objects, while Slot Attention consistently separates objects even in the presence of textures. In Fig. 13 (Appendix G), we show similar results on CLEVR. See also Fig. 12 for results on additional images from Multi-dSprites.

Figure 4: Median performance of the different seeds trained for each of the indicated models (error bars denote 95% confidence intervals). Top: ARI (↑) and MSE (↓) for each dataset and model. Bottom: performance of the downstream model on each feature of the objects. Accuracy is used for categorical features and $R^2$ for numerical features. A random guess baseline is shown in purple.

refer to Appendix E for further details and results on the hyperparameter search for MONet and Slot Attention.

When qualitatively looking at the results obtained by the hyperparameter search performed for MONet, we notice a consistent inability of the "component VAE" to capture different characteristics other than simple colors (see Fig. 3), even when strongly penalizing the VAE for reconstructing shapes that are inconsistent with the masks computed by the recurrent attention network (which are directly used for reconstruction). Here, we show and discuss quantitative and qualitative results for the combination of hyperparameters that achieves the best performance, which we call *best model*. We now discuss the results obtained using the combination of hyperparameters that achieves the best performance, called *best model*. In Fig. 2b, we see that MONet still hyper-segments even though the reconstructions are now blurred. For Slot Attention (Fig. 2d), we observe that the quality of reconstructions has improved, and it more often represents an entire object in a single slot. Although the ARI for MONet has also improved, the separation problem is still far from solved, while Slot Attention shows a significant improvement both in terms of ARI and MSE (see two uppermost plots in Fig. 4). Note how, for Slot Attention, the ARI is significantly lower in Multi-dSprites, when compared to CLEVR. The likely reason is that, when a significant portion of an object is occluded by another, the visible shape is being altered significantly and the edges of objects are not clear. Therefore, two explanations of the same scene can

still be reasonable while not corresponding to the ground truth. This extreme overlap never occurs in CLEVR.

Overall, even when MONet sacrifices reconstruction quality and blurs away the details, hyper-segmentation is still present as evidenced by our qualitative and quantitative analyses. This suggests that the separation problem in MONet may not simply be caused by the training objective, but rather by its architectural biases. Indeed, improving the reconstruction performance of Slot Attention has, instead, yielded both better separation and more detailed reconstructions, suggesting that generating shape and appearance using a single module is a more favorable inductive bias for learning representations of objects with complex textures.

**Representation performance.** To understand the interplay between separation and learned representations, we explore the performance on a downstream property prediction task that was trained using the object representations as inputs (see Appendix C.1 for details). Only the properties that were not affected by the change in texture are considered, to ensure a fair assessment of the quality of the representations. From Fig. 4, we observe how MONet fails to capture some of the properties in the representations and consistently performs worse than Slot Attention, for both the baseline and the improved versions. This suggests that, as highlighted in Dittadi et al. (2021b), a model that is not capable of correctly separating objects will also fail to accurately represent them.

The trend is also clear from Fig. 5, which shows that a higher ARI score strongly correlates with an increased performance of the downstream model on all object properties. The correlation with MSE is much weaker, which highlights

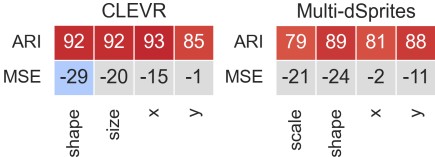

Figure 5: Rank correlation of the ARI and MSE scores with downstream property prediction performance. Correlations are computed over *all* the models trained with that dataset in our study.

how *strong visual reconstruction performance is not the ultimate indicator for good object representations*. This result does not contradict previous findings (Dittadi et al., 2021b) as here the properties we expect the downstream model to predict have little to do with the texture of the object and, therefore, the model can have poorer reconstructions while still obtaining useful representations.

**Representation bottleneck.** Object representations are typically obtained by using a low-dimensional latent space for each object. This constitutes a bottleneck in the model, which we term *representation bottleneck* (see Appendix A). Here, we investigate how the size of this bottleneck affects a model's performance.

We observe from Fig. 6 that the MSE improves for both MONet and Slot Attention when the latent space size increases. However, for MONet this comes with a decrease in separation performance. For Slot Attention, when the latent space reaches a critical size (256 in CLEVR and 512 in Multi-dSprites), the performance degrades, and the variability across seeds increases drastically, suggesting this may be due to optimization problems during training. The increase in latent space size arguably increases the model's capacity, but it does not prove to be enough to significantly improve separation and reconstruction. Instead, we can obtain considerable improvements only by changing architecture.

## 4  RELATED WORK

Learning representations that reflect the underlying structure of data is believed to be useful for downstream learning and systematic generalization Bengio et al. (2013); Greff et al. (2020); Lake et al. (2017). While many recent empirical studies have investigated the usefulness of disentangled representations and the inductive biases involved in learning them Dittadi et al. (2021c, 2022); Locatello et al. (2019); Montero et al. (2021); Träuble et al. (2021); Van Steenkiste et al. (2019), analogous experimental studies in the context of object-centric representations are scarce. The study by Engelcke et al. (2020a) presents an investigation into inductive biases for object separation, focusing on one model and traditional synthetic datasets. In this work, we study hyper-segmentation on datasets where objects have complex

textures. Karazija et al. (2021) recently proposed ClevrTex, a dataset that introduces challenging textures to scenes from CLEVR (Johnson et al., 2017). The experiments reported show that, without any tuning, some models fail to segment complex scenes by focusing on colors. The authors, similarly to what we highlighted in this work, state that "ignoring confounding aspects of the scene rather than representing them might aid in the overall task [of segmentation]." In our work, we investigate the mechanism behind the ability of some models to ignore superfluous details and more successfully segment the image, proposing a useful inductive bias to achieve better object representations.

Recently, works that propose new object-centric learning methods also include evaluations on more complex datasets. Greff et al. (2019) train IODINE on Textured MNIST and ImageNet, and observe that the model separates the image primarily according to color when the input is complex. GENESIS-V2 (Engelcke et al., 2021) was trained on Sketchy and APC, two real-world robot manipulation datasets. However, the authors do not explore the mechanism behind the performance of the models they tested, and do not attempt to optimize the architectures. In the video domain, Kipf et al. (2021) include evaluations on a dataset with complex textures, training their model to predict optical flow rather than a reconstruction of the input.

Relevant work naturally includes object-centric learning methods. Here, we provide a brief overview focusing on slot-based models—a subset of object-centric models that includes those studied in this paper. We will be using the categorization proposed by Greff et al. (2020) to distinguish different ways to tackle object separation.

Models based on **instance slots** (Chen et al., 2019; Goyal et al., 2019; Greff et al., 2016, 2017, 2019; Huang et al., 2020; Kipf et al., 2019, 2021; Le Roux et al., 2011; Locatello et al., 2020; Löwe et al., 2020; Racah and Chandar, 2020; Van Steenkiste et al., 2018; van Steenkiste et al., 2020; Yang et al., 2020) represent objects with individual slots that share a common format of representation, making no assumption regarding interaction between objects. In short, this introduces a routing problem, as the slots are all equally capable of representing a given object present in the input. To solve this ambiguity, an explicit separation step needs to be introduced, often utilizing some form of interaction between slots.

Models that use **sequential slots** (Burgess et al., 2019; Engelcke et al., 2020b, 2021; Eslami et al., 2016; Kosiorek et al., 2018; Kossen et al., 2019; Stelzner et al., 2019; von Kügelgen et al., 2020) implement the same shared format between slots as instance slots based models, but here the routing problem is solved by reconstructing each object sequentially, conditioning on previous ones. The lack of independence between slots can lead to a decrease in the compositionality of the representations.

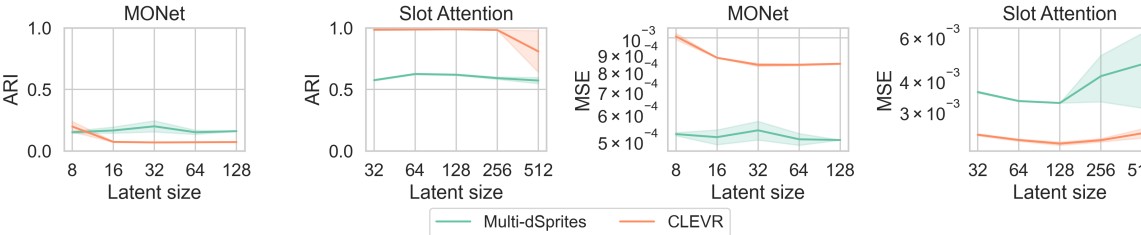

Figure 6: The ARI (↑) and MSE (↓) results for different latent space sizes show that the *representation bottleneck* (see main text) is not a sufficient inductive bias for object separation, as there is no significant change in the ARI metric. Slot Attention is prone to training instability when the latent size exceeds 256. We use two seeds for each latent dimension, and show mean (line) and 95% confidence interval (shaded area).

Finally, models based on **spatial slots** (Crawford and Pineau, 2019, 2020; Deng et al., 2021; Dittadi and Winther, 2019; Jiang et al., 2020; Lin et al., 2020a,b; Nash et al., 2017) associate each slot with a spatial coordinate, providing more explicit positional bias in the representation and reconstruction process.

# 5    CONCLUSIONS

In this paper, we have investigated which inductive biases may be most useful for slot-based unsupervised models to obtain good object-centric representations of scenes where objects have complex textures. We found that using a single module to reconstruct both shape and visual appearance of objects naturally balances the importance of these two aspects in the generation process, thereby avoiding hyper-segmentation and achieving a better compromise between precise texture reconstructions and correct object segmentation. Therefore, our recommendation is that models should have separation as an integral part of the representation process. Additionally, we showed that separation strongly correlates with the quality of the representations, while reconstruction accuracy does not: this justifies sacrificing some reconstruction quality. Finally, we observed that the representation bottleneck is not a sufficient inductive bias, as increasing the latent space size can be counterproductive unless the model is already separating the input correctly.

We limited our study to two models based on instance slots and sequential slots. Although the models considered in our study have been shown to be among the most successful ones on this type of data, it would be interesting and natural to extend our study and explore if the same holds for other models that approach the problem in a similar way, such as GENESIS, IODINE, and GENESIS-V2, as well as methods based on spatial slots, such as SPAIR or SPACE.

Another interesting avenue for future work is to extend our study to more complex downstream tasks involving abstract reasoning, e.g., in a neuro-symbolic system, where symbol manipulation can be performed either within a connectionist framework (Battaglia et al., 2018; Evans and Grefenstette, 2018; Smolensky, 1990) or by purely symbolic methods (Asai and Fukunaga, 2018; Dittadi et al., 2021a; Mao et al., 2019). Finally, it would be relevant to validate our conclusions on additional datasets, and to introduce objects with varying texture complexity, as this could require different model capacities to achieve separation (Engelcke et al., 2020a).

## Acknowledgements

We would like to thank Francesco Locatello and Bernhard Schölkopf for valuable discussions and feedback.

Work partially done while Samuele Papa was a MSc student at the Technical University of Denmark.

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

# A    REMARKS ON NOTATION

The term *representation bottleneck* should not be confused with the *reconstruction bottleneck* introduced by Engelcke et al. (2020a). The representation bottleneck refers to the small size of the latent space, while the reconstruction bottleneck refers to how easy it is for the model to reconstruct the data. In Engelcke et al. (2020a), the reconstruction bottleneck is posited to be the reason behind the models' inability to separate objects into different slots.

Often, in the paper, we refer to *object-centric representations* and *slots* as synonyms, although this is only true for slot-based models.

The term *hyper-segmentation* refers to when a model splits the input into different slots with little to no regard to high-level characteristics of the input, such as the shapes of objects, and instead just uses low-level characteristics, primarily color. This often results in slots that consist of small clusters of pixels with similar color, which means that several objects can be partially represented in the same slot and at the same time each object may be partially represented in multiple slots. This phenomenon is distinct from *over-segmentation* (Engelcke et al., 2020b), where multiple slots may reconstruct a single object but no slot reconstructs (parts of) multiple objects. Examples are shown in the main text of the paper (see Fig. 2b and Fig. 2b), where MONet is hyper-segmenting the input, while Slot Attention is sometimes over-segmenting it.

# B    DATASETS

The original versions of both datasets are taken from Kabra et al. (2019).

**CLEVR.** The CLEVR dataset consists of 3D objects placed on a gray background at different distances from the camera. Overlap between objects is kept to a minimum. There are spheres, cylinders, and cubes of eight different colors. The objects can be metallic of opaque. There is a big and small variant of each object and they can be placed in several different orientations. We use the variant of the dataset that has no more than 6 objects in it, as was done in previous object-centric learning research. The total number of samples in the training dataset is $49483$, and we leave 2000 samples for validation and 2000 samples for testing.

**Multi-dSprites.** The Multi-dSprites dataset places several 2D objects on a grayscale background. The objects can be a square, an ellipse, or a heart. They can have any RGB color, orientation, and different levels of overlap. Here, we use 90000 samples for the training, 5000 for validation, and 5000 for testing.

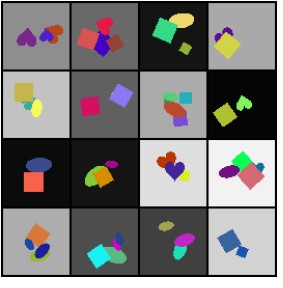 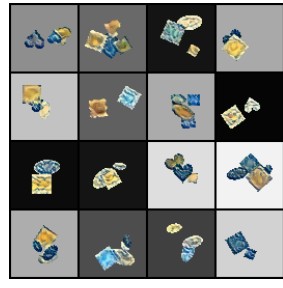

(a) Samples from original Multi-dSprites.    (b) Samples from style transfer Multi-dSprites.

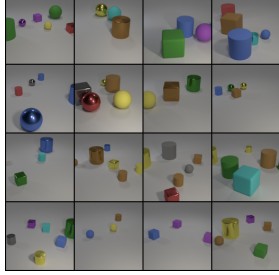 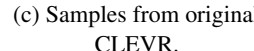 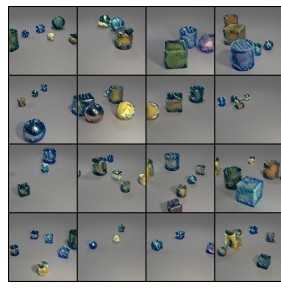

(c) Samples from original CLEVR.    (d) Samples from style transfer CLEVR.

Figure 7: Samples from the original and neural style transfer datasets.

**Neural Style Transfer.** Neural Style Transfer was applied in its most basic form (Jacq, 2021) except for a few additions to make running it on several datasets easier. We opted to use *The Starry Night* by Dutch painter *Vincent Van Gogh* as a reference style image (we used the photo from Wikimedia Commons, which is in the public domain). We experimented with several parameters, and we noticed a lot of variability between runs and a more pleasant result from the most basic implementation of the algorithm.

The final version of the datasets was obtained by first applying neural style transfer to each image (optimization happens on an image-to-image basis). This results in the entire scene having the style of the reference image. After obtaining the neural style transfer version of the image, we applied the original segmentation masks of the objects to obtain an image where only the foreground objects have a complex texture, while the background remains the original one.

# C EVALUATION

## C.1 DOWNSTREAM FEATURE PREDICTION TASK

The setup for the feature prediction task is the same as the one used in Dittadi et al. (2021b). The models used are a simple linear model and an MLP with one hidden layer having 256 neurons and enough outputs to predict all of the features of an object. The input to the model is the object representation of a single object and the output is the predicted features for that object. Let $r$ be the representation of an object, $M$ the model, $\hat{y} = M(r)$ is the output of the model, and $y$ is the target vector such that $y_{i_k:i_{k+1}}$ is the $k$th feature of the object, a vector of dimension $(i_{k+1} - i_k + 1)$. We use the MSE loss for numerical features and the cross-entropy loss for categorical ones.

Now, it is important to note that, in order to correctly train the model, the representation $r$ needs to be matched with the target vector $y$ of the object that $r$ is representing. However, this is very challenging, as the models can represent any object in any of the slots. Therefore, following Dittadi et al. (2021b), we adopted two different strategies to match slots with objects. The first is called *loss matching*: The loss for each pair of slot and object is computed, resulting in a loss matrix $L$, where $L_{i,j}$ is the loss between the predicted features from the $j$th slot and the target features from the $i$th object in the scene. Then, the Hungarian matching algorithm is used to find the pairs of slots-objects that minimize the sum of the loss. The second approach is called *mask matching*: The masks predicted by the models and the ground masks are matched, to find the pairs that have the smallest difference. By using loss matching, the assumption is that the initial errors that are inevitable (because the downstream model has not been trained yet) will eventually disappear. When using mask matching, this problem disappears, however, we rely on the ability of the models to generate masks that closely match the ground truth, which is not the case for models that are hyper-segmenting the input, as is often the case in our study.

## C.2 ARI AND MSE

We use the standard definitions of Adjusted Rand Index and Mean Square Error.

The ARI is computed on the foreground objects and is meant to measure the similarity between two partitions of the same set. The *adjusted* part of the name comes from the fact that the Rand Index has been normalized according to a null hypothesis to give $0$ when the partitions are random and $1$ when they coincide.

The MSE is computed between each channel in each pixel of the image.

# D IMPLEMENTATION OF THE MODELS

The models were re-implemented in PyTorch (Paszke et al., 2019) and run on NVIDIA A100 or NVIDIA TitanRTX GPUs. The total approximate training time to reproduce this study is 300 GPU days.

# E HYPERPARAMETER SEARCHES

## E.1 BASELINES

The baselines were obtained by training the models on the two datasets with 3 different seeds. The parameters are taken from the original papers, but for MONet we use different values for the foreground and background sigma, as suggested by Greff et al. (2019). We stopped the training for all runs in our study, even the ones described later, to 500k steps.

| Parameter | Value(s) |
|---|---|
| foreground sigma | 0.05, 0.5 |
| background sigma | 0.03, 0.3 |
| gamma | 0.05, 1, 5 |
| latent size | 64 |
| latent space MLP size | 128 |
| decoder input channels | 66 |
| number of skip-connections in U-Net | 0, 3, 5 |
| dataset | CLEVR, Multi-dSprites |

Table 1: Hyperparameter search for MONet.

## E.2 IMPROVING MONET

Starting from the baseline results, we first explored the hyperparameter space manually, to develop an intuition regarding the effect of each hyperparameter on the performance.

We then performed a hyperparameter search for MONet. We ran a full search, resulting in 36 runs. Because of the high number of runs, we decided to use a single seed. The parameters are listed in Table 1. Those that are not listed were kept unchanged. All combinations of parameters are tested, but foreground sigma and background sigma have been changed in pairs, so that when the foreground sigma is $0.05$, the background sigma is $0.03$ and when foreground sigma is $0.5$, background sigma is $0.3$ to keep consistent weights of the reconstruction loss.

Some analysis on the results of these models can be seen in Fig. 8, where we can see how the parameters have little to no impact on the overall performance of the model. What

proved to be most effective was reducing the number of skip connections in the U-Net and using a small sigma for the loss function. However, these results are not very conclusive, as a small number of skip connections is actually just increasing the ARI slightly by reconstructing bigger patches of objects in the slots and not actually separating them correctly.

### E.3 REPRESENTATION BOTTLENECK STUDY

The representation bottleneck study was done by changing the latent space of both MONet and Slot Attention with 2 seeds and without changing any other parameter, resulting in 32 runs. The latent sizes tested are shown in Table 2. The findings are summarized in the main text of the paper.

| MONet | Slot Attention |
|---|---|
| 8 | 32 |
| 16 | 64 |
| 32 | 128 |
| 64 | 256 |
| 128 | 512 |

Table 2: Latent space sizes tested in the study for each of the two models.

### E.4 IMPROVING SLOT ATTENTION

We tried to increase the size of the encoder and decoder architecture as much as possible, while being reasonable regarding training time and GPU memory. We tested several architectures, with the objective of improving the overall reconstruction quality by reducing the blurriness. We quickly realised that we needed a very deep architecture, therefore, we opted to use residual layers. The final architecture managed to achieve the best results when averaged over 3 different seeds. Each layer is a stack of two convolutional layers, with Leaky ReLU activation functions, a skip connection and we also employ the re-zero strategy (Bachlechner et al., 2021). We increased the latent size to 512, used upscaling in the encoder and downscaling in the decoder. We fixed the broadcast size of the decoder to 32. We used a stack of 16 residual blocks. The architecture of the encoder is described in Table 3, and the decoder is symmetrical (starting from 256 channels going down and instead of downscaling we have upscaling). To map from the input number of channels to the desired ones we use an additional convolutional layer, the same for the output channels. We did not experiment with the number of iterations that the Slot Attention Module performs, but it would be interesting to understand whether this parameter is very influential in natural scenes.

| Name | Number of channels | Activation / Comment |
|---|---|---|
| Residual Block | 64 | Leaky ReLU |
| Residual Block | 64 | Leaky ReLU |
| Residual Block | 64 | Leaky ReLU |
| Residual Block | 64 | Leaky ReLU |
| Downscaling | | Only for CLEVR |
| Residual Block | 64 | Leaky ReLU |
| Residual Block | 64 | Leaky ReLU |
| Residual Block | 64 | Leaky ReLU |
| Residual Block | 64 | Leaky ReLU |
| Downscaling | | |
| Residual Block | 128 | Leaky ReLU |
| Residual Block | 128 | Leaky ReLU |
| Residual Block | 128 | Leaky ReLU |
| Residual Block | 128 | Leaky ReLU |
| Residual Block | 256 | Leaky ReLU |
| Residual Block | 256 | Leaky ReLU |
| Residual Block | 256 | Leaky ReLU |
| Residual Block | 256 | Leaky ReLU |

Table 3: Final encoder used for the Slot Attention model that obtained the best results in terms of both ARI and MSE. Residual Blocks always have 2 convolutions and use ReZero (Bachlechner et al., 2021), two downscaling operations are used for the CLEVR dataset, while one for Multi-dSprites. The decoder is perfectly simmetrical to this structure.

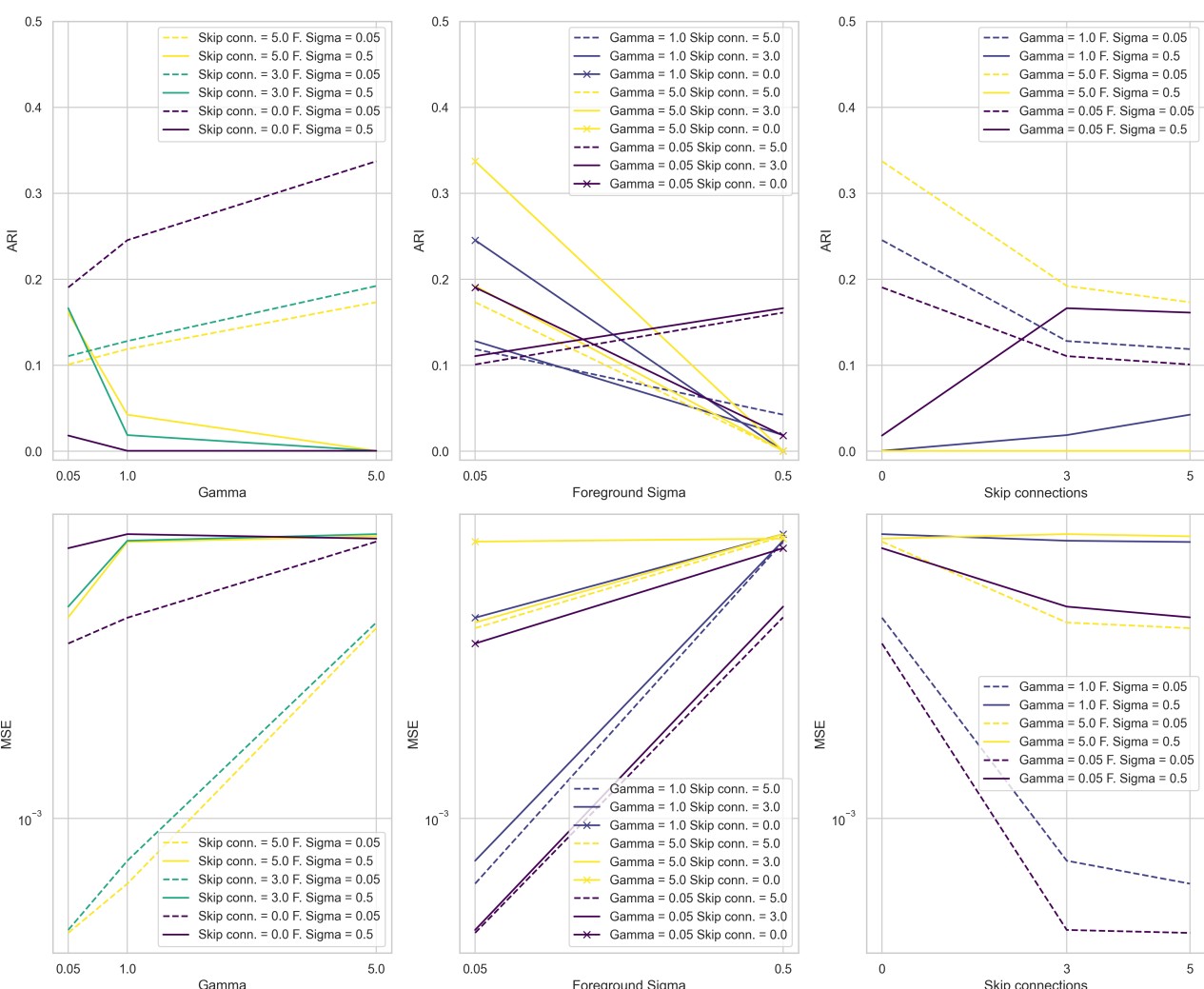

Figure 8: Results (top row: ARI; bottom row: MSE) from the hyperparameter search for MONet. Although increasing gamma or foreground sigma in the loss function successfully deteriorates reconstruction performance (MSE), they are not sufficient to improve the ARI (in fact, the increase of foreground sigma actually decreases the ARI). A smaller number of skip connections also achieves the desired higher MSE, which corresponds to higher ARI only for small values of sigma. Often, having big values for gamma and sigma results in trends opposing the desired ones in terms of ARI.

# F  ADDITIONAL RESULTS

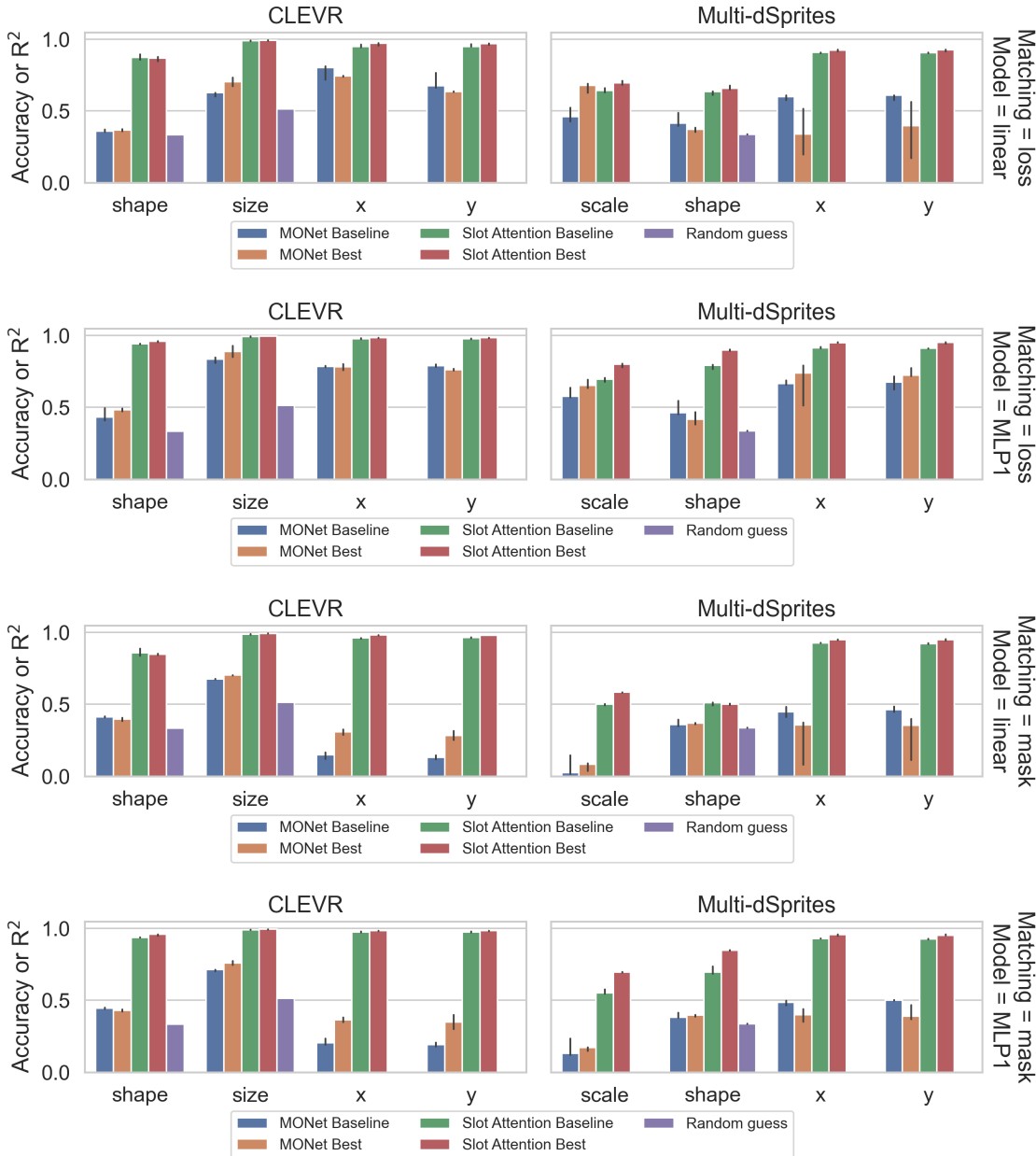

Figure 9: Comparison of the downstream performance for all combinations of slot-object matching and model type (results on from the test set, downstream models trained on the validation set). We notice how accuracy (↑) and $R^2$ (↑) both increase significantly for both MONet models when using loss matching compared to mask matching (especially for numerical features). This is expected, as the masks generated by MONet suffer substantially from hypersegmentation, which makes mask-matching a very unstable way to pair slots with the correct object. Instead, Slot Attention manages to generate more accurate masks, which results in more consistent performance between mask and loss matching methods.

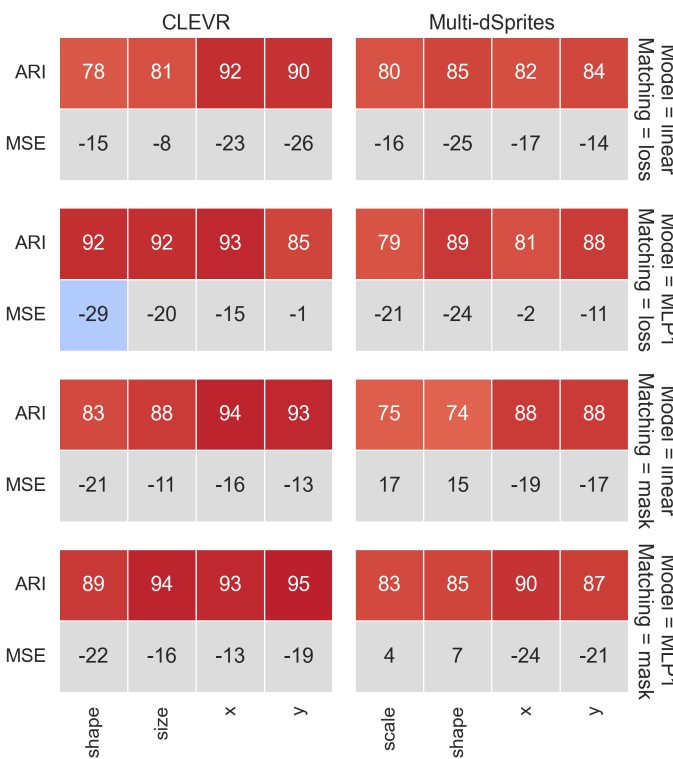

Figure 10: Pearson's correlation coefficient for all runs, grouped by dataset and showing the different combinations of matching and downstream model. The correlation between downstream model's performance and the ARI (↑) and MSE (↓) metrics shows that ARI is a strongest indicator of good representation quality when the object-centric models are being trained on data with complex texture. Difference in correlation between different matching methods and downstream models is again to be attributed to the poor mask generation quality, which makes mask matching very challenging.

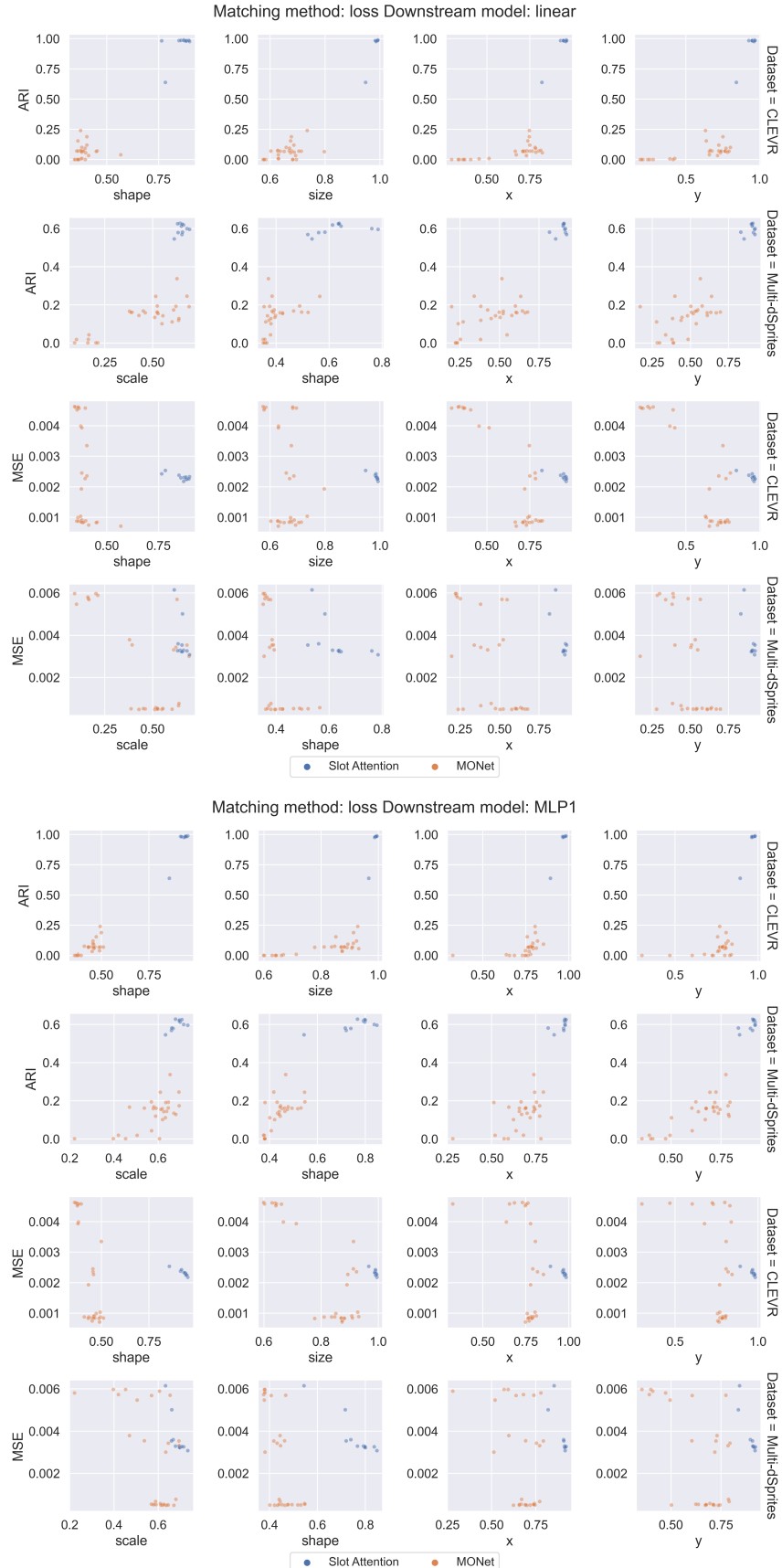

Figure 11: Scatter plots to inspect correlation between downstream performance, and ARI (↑) and MSE (↓). The color shows the different models, which clearly display distinct patterns. A visual inspection shows that there is very little correlation between downstream performance and MSE. Only loss matching is shown here.

# G   QUALITATIVE RESULTS

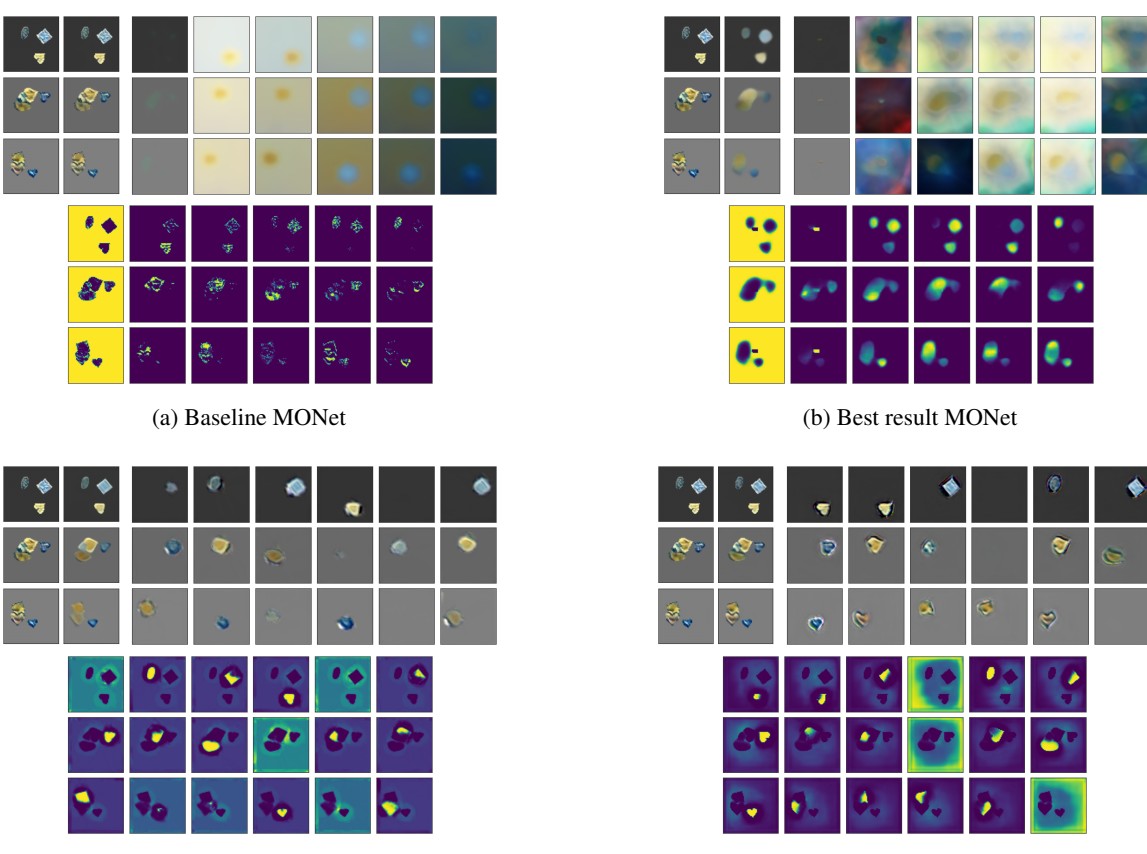

(a) Baseline MONet

(b) Best result MONet

(c) Baseline Slot Attention

(d) Best result Slot Attention

Figure 12: Qualitative results for the separation performance of the models in the comparative study on *Multi-dSprites*. From left to right in all subfigures: (top) input, final reconstruction, reconstruction for each of the six slots (no predicted mask is applied here, only the visual appearance part of the reconstruction is shown), (bottom) mask for each of the six slots. Here the difference between the two versions of Slot Attention is even more noticeable, and we can see how MONet is blurring the masks. However, MONet never manages to reconstruct the correct visual appearance, even when a more accurate shape of the objects is being predicted by the attention module. Balancing visual appearance and shape is much more challenging in MONet.

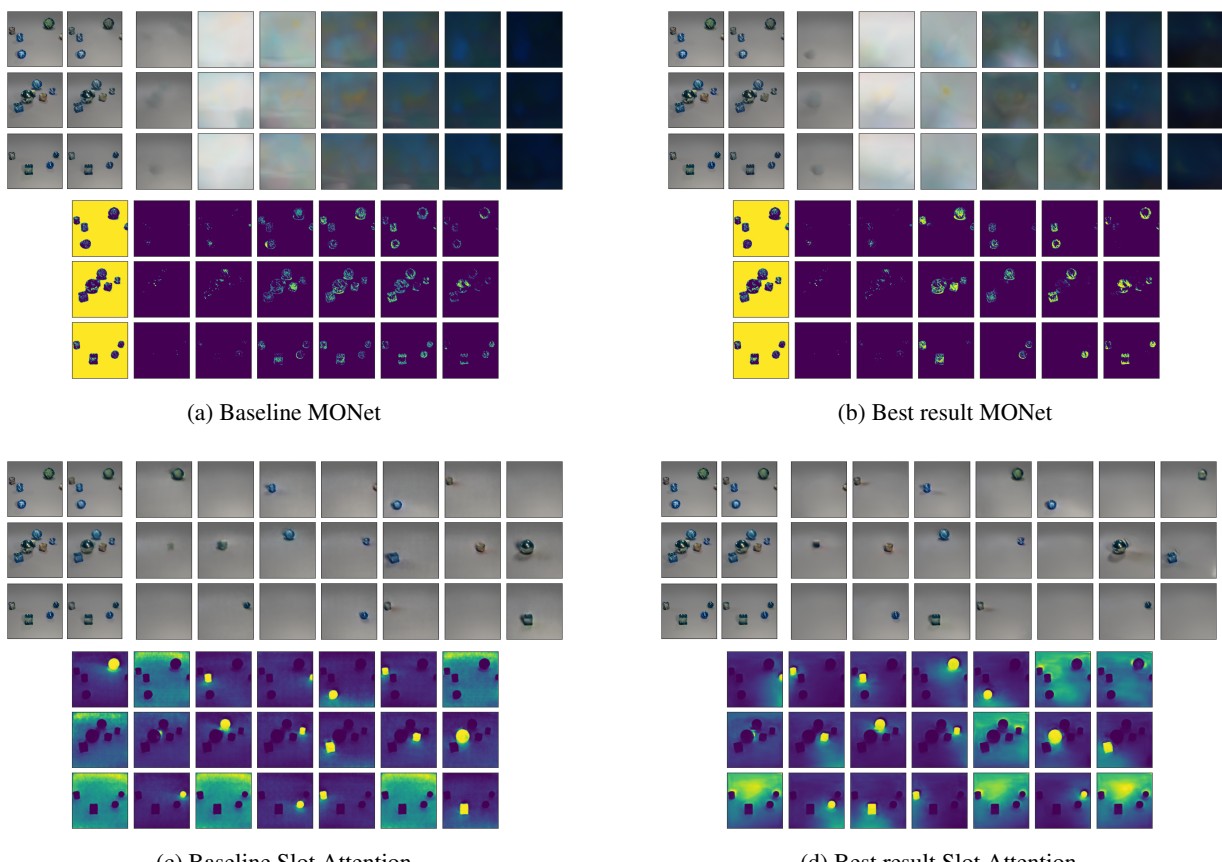

(a) Baseline MONet

(b) Best result MONet

(c) Baseline Slot Attention

(d) Best result Slot Attention

Figure 13: Qualitative results for the separation performance of the models in the comparative study on *CLEVR*. From left to right in all subfigures: (top) input, final reconstruction, reconstruction for each of the six slots (no predicted mask is applied here, only the visual appearance part of the reconstruction is shown), (bottom) mask for each of the six slots. The masks on the improved Slot Attention start to include more of the background for each object. In both baseline and best result, Slot Attention isolates each object in a distinct slot, rarely over-segmenting the input, a stark difference when comparing to Multi-dSprites. For MONet, it manages to perform better in CLEVR than Multi-dSprites, however, the best result is still hypersegmenting the input and not blurring it. Overall, MONet cannot reconstruct the visual appearance using the VAE, and leaves all the heavy lifting to the attention module.