# OpenReview forum: "Inductive Biases for Object-Centric Representations in the Presence of Complex Textures"
_auai.org/UAI/2022/Workshop/CRL — CRL@UAI 2022 Poster_

### Official Review · Reviewer_GYEA · 2022-06-28
**Review: Object-Centric Representations with Complex Textures**

**Rating:** 6
**Confidence:** 3

**Review:**

### Summary
This paper uses neural style transfer to create textured object datasets and measures the performance of MONet and Slot Attention on these datasets. The authors find that the inductive bias of MONet causes the model to rely primarily on color information via what the authors deem hyper-segmentation while Slot Attention relies on shape information and is thus more robust. The authors also show that the segmentation achieved by a model correlates with the quality of the learned representation and that increasing the latent capacity does not help in segmenting objects.


### Pros
* The paper is well written and the main contributions are presented clearly
* The paper tackles the highly relevant question of understanding the mechanisms underlying the success and failure of object-centric models on realistic dataset
* The findings wrt hyper-segmentation in MONet relative to Slot Attention are interesting and convincingly presented both qualitatively and quantitatively.
*  The authors concisely present clear recommendations based on their results to the community.

### Cons
My main issue with this work has to do with the novelty and new insight offered by the experimental findings of the authors relative to prior work

* As the authors mention, the creation of textured object datasets has been done before, and I didn’t feel that the authors sufficiently justified why new datasets were needed opposed to just using e.g. ClevrTex

*  The conclusion that there is not a strong relationship between reconstruction accuracy and representation quality in object-centric models seems a bit trivial to state. If MSE were the ultimate metric then we could just use vanilla auto-encoders.

* Regarding the experiments on increasing the latent capacity, it has been shown in prior work that using a large latent capacity does not benefit segmentation performance [1]. I suppose the novelty here is that the dataset considered contains more realistic textures, however, I don't find that this yields any particularly novel insight.

### Conclusion
Despite my issues with some of the insights offered in this paper, I do think that the authors tackle a very relevant question and provide some interesting insights wrt the discrepancies in the inductive biases of MONet and Slot Attention. I am borderline, however, on whether these contributions are enough to justify acceptance. To this end, I am giving a weak accept.

### References
[1] https://arxiv.org/pdf/2007.06245.pdf

---

### Official Review · Reviewer_dcXF · 2022-07-01
**new empirical findings, but immature work**

**Rating:** 6
**Confidence:** 4

**Review:**

This work conducted a systematic experimental study on the inductive biases for object-centric representations when objects have complex textures. Specifically, current object-centric learning approaches typically tested on relatively simple datasets. This work use neural style transfer to apply complex textures to the objects and use the synthetic data to test existing methods’ representation performance.

Pros:
The idea of investigating the representations of objects with complex textures rather than synthesis simple data is impressive.

Concerns:
1.  There are two methods tested in this paper, MONet and Slot Attention. I think only two methods might not represent all the disentangled or non-disentangled methods and thus draw the conclusion that the latter is better. It would be better to take more methods into consideration.
 2. This work only adopts The Starry Night as the reference style image which is still far from natural objects. What happens if using other more neutral images?
3. The downstream tasks are all about object detections, which is hard to adequately reflects the representation performance due to the lack of texture based tasks.

Overall,  this paper contains some novel ideas, but the experiments is too simplistic to verify the new ideas.

---

### Meta-Review · Program_Chairs · 2022-07-06

**Recommendation:** Accept (Poster)
**Confidence:** 3

**Metareview:**

While reviewers raised different concerns on this work, the question it tackles was deemed relevant and the the contributions concise and clear. For a future version, authors are encouraged to stress the significance of their results in the context of the CRL workshop topics and focus.

---

### Decision · Program_Chairs · 2022-07-06

Accept (Poster)